# THE WISDOM OF THE CROWD: RELIABLE DEEP REINFORCEMENT LEARNING THROUGH ENSEMBLES OF Q-FUNCTIONS

## ABSTRACT

Reinforcement learning agents learn by exploring the environment and then exploiting what they have learned. This frees the human trainers from having to know the preferred action or intrinsic value of each encountered state. The cost of this freedom is reinforcement learning is slower and more unstable than supervised learning. We explore the possibility that ensemble methods can remedy these shortcomings and do so by investigating a novel technique which harnesses the wisdom of the crowds by bagging Q-function approximator estimates.

Our results show that this proposed approach improves all three tasks and reinforcement learning approaches attempted. We are able to demonstrate that this is a direct result of the increased stability of the action portion of the state-action-value function used by Q-learning to select actions and by policy gradient methods to train the policy.

## 1 INTRODUCTION

In the reinforcement learning (RL) approach the agent learns by exploring its environment and the, sometimes, many approaches to solving a given problem. It frees the human trainers from having to know the preferred action or intrinsic value of each encountered state. There is no denying that RL has a grassroots feel: it is an important form of learning in the natural world and it is only natural that machine learning practitioners would want to mimic its success.

This freedom comes at a price, however. The most common complaints about RL, especially when using function approximators to learn Q-functions, are that RL is too slow and unstable during learning. Learning by exploration of the environment results in a training signal which is less informative than a supervised training signal resulting in the requirement of a large number of training samples and repeated exposure to those samples. The proposed approach is an ensemble learning approach to RL similar to the well-known bagging approach, Breiman (1996), which trains ensemble members using the experience replay memory and combines their action selections using voting or averaging.

### 1.1 RECENT ADVANCEMENTS IN RL

There has been tremendous advancements in RL in recent years spurred by the excitement surrounding deep Q-learning (DQN), Mnih et al. (2015). Impressively, RL has featured prominently in published work showing super-human performance in tasks which were previously considered untouchable by state of the art RL approaches just a few years ago. These include RL agents playing chess at a grand-master level, Lai (2015), and the famous AlphaGo, Silver et al. (2016). Through this work we have found just how far we can go when applying existing RL approaches with large amounts of computing power.

A large effort has been devoted to addressing the RL obstacle of slow learning, Mnih et al. (2016); Schaul et al. (2016), with an emphasis on speeding-up RL for the high-dimension inputs popularized by the DQN work – especially since DQN exacerbated this issue both in terms of the required computation and number of training time steps. There has also been work showing other RL approaches can be adapted to the DQN paradigm, Lillicrap et al. (2016).

For all of these recent works mentioned here the primary emphasis is achieving improved performance on simulated tasks in the shortest amount of wall-clock time. The issues of RL instability during training and of reducing the number of interactions with the training environment have received little attention. The crowd ensemble addresses the obstacle of training instability without sacrificing the number of interactions with the training environment which, in many real-world application, may be more expensive than the computational costs. Furthermore, the crowd ensemble method can be used alongside any of these recent advances in RL.

## 1.2 THE WISDOM OF CROWDS

Francis Galton, in 1906, observed that a large group's mean guess was able to come within one pound when guessing the weight of an Ox, Galton (1907). This was surprising because the crowd, while containing a few potential experts, was presumably made of non-experts with no knowledge of estimating the weight of oxen.

More recently Treynor (1987) emphasizes the need for independence of the individual to the success of the crowd and that allowing sources of shared error or bias will reduce the accuracy of a crowd's prediction. He concludes that chasing an expert is folly and that a simple combination of crowd information is best. Larrick & Soll (2006) and Hastie & Kameda (2005) emphasize the power of averaging predictions. Appropriate use of averaging in group decisions begins with understanding that averaging is not a regression to the mean but, rather, an error-reduction technique

Surowiecki (2005) synthesizes the many benefits of crowd-based decision-making. Although the crowd will not regularly out-perform the best individual, no field can provide a mechanism to predict which individual will out-perform the crowd. Fortunately, many of the obstacles to quality crowd decision-making are a result of the shortcomings and complexities of human group dynamics are not problems we encounter when training Q-function ensembles.

## 2 RELATED WORK

Most ensemble RL methods fit the mixture of experts paradigm described in Jacobs et al. (1991). A Gaussian mixture model approach is utilized as the ensemble mechanism by Agostini & Celaya (2011). This gives each ensemble member a region of expertise. They test their approach on the pendulum swing-up and cart-pole balancing tasks. An couple ensembles of ANN Q-function approximators are compared in a limited set of experiments on the pole-balancing task in Hans & Udluft (2010). They conclude that a hard combination of ensemble members, that is only one expert is active at a time, is superior to the soft-combination of experts they initially attempted.

In the multiple model reinforcement learning (MMRL), Doya & Samejima (2002), approach each expert has a forward model and a Q-function. The forward models determine how to combine member outputs and how to backpropagate the error signals. In Doya & Samejima (2002) MMRL is tested on the non-stationary pendulum swing-up task. For these experiments they are forced manually partition the state space.

In Faußer & Schwenker (2015) a concept very similar to the crowd ensemble method is proposed. Their approach has an important difference: the ensemble members are trained in parallel with each other. This results in a factor of $N_e$ more interactions with the training environment. Their results show that their approach with $N_e > 3$ performs better than a single Q-learner on a maze navigation task and a simplified tetris task (SZ-tetris) with a reduced set of pieces. No explanation is given for the improved performance of the ensemble.

Recently Q-learning approaches have been developed which resemble ensemble methods and may share some of the same benefits. These approaches are not ensembles because they do not have a mechanism which combines ensemble member outputs.

Mnih et al. (2016) and Nair & Silver (2015) present a RL approach which uses multiple, simultaneous simulations to speed-up DQN. In Mnih et al. (2016) this method is called asynchronous DQN. True to its name, the parameter updates are not synchronized allowing each simulation to periodically update the global parameter values using an accumulated gradient over several time steps. Asynchronous DQN is tested using a large number of arcade learning environment.

Double Q-learning, Hasselt (2010). is designed to address the issue of over estimation of Q-values. This is done by leveraging two Q-functions. In a later work the double Q-learning concept is adapted to the DQN framework van Hasselt et al. (2015). The double Q-learning concept is taken a step further by investigating using any number of Q-learners which they call multi Q-learning, Duryea (2016).

## 3 METHOD

Here we briefly describe the crowd ensemble approach to Q-learning and the experiments used to evaluated the proposed method. We begin with the crowd ensemble.

### 3.1 THE CROWD ENSEMBLE APPROACH TO Q-LEARNING

Q-learning, Sutton & Barto (1998), learns a state-action function, $Q(s, a)$ which is updated according to:

$$Q(s(t), a(t)) \leftarrow Q(s(t), a(t)) + \alpha \big[ r(t+1) + \gamma \max_{a'} (Q(s(t+1), a')) - Q(s(t), a(t)) \big] \quad (1)$$

where $t$ is the current time step, $Q(s(t), a(t))$ is the Q-function value for state/action pair $(s(t), a(t))$, $s(t+1)$ is the state which occurs when taking action $a(t)$ from state $s(t)$, $r(t)$ is the reward at time $t$, $\alpha$ is a learning rate parameter, and $\gamma$ is a parameter modulating the effect of future rewards. The value $\max_{a'}(Q(s(t+1), a'))$ is the current estimate of future cumulative reward: it represents the current belief of the best possible action, $a'$, taken at the next state, $s(t+1)$.

Using a feed-forward artificial neural network (ANN) to model a Q-function is a common function approximation method when applying Q-learning to tasks with continuous state spaces, Anderson et al. (2015). This is typically done by having a single ANN output represent $Q(s, a)$ and inputting $s$ and $a$ into the network. Updates are performed by backpropagating the error computed in (1) through the ANN. Alternatively, as is done in the DQN approach, the ANN can have a multiple outputs, one for each $Q(s|a)$, the value of $Q(s, a)$ given an action, with only the state as inputs.

Prior to learning, the Q-function approximator parameters for each expert should be randomly initialized. Learning in a crowd ensemble follows these steps.

1. Vote: select an action for each expert.
2. Tally: determine which action is selected by the most members.
3. Act: take the action selected by the ensemble.
4. Observe: store the new state and store it in the experience replay memory.
5. Sample: independently sample from the experience replay memory for each member.
6. Compute errors: compute a TD-error for each member.
7. Update: update Q-function approximations using each member's TD-error.

Our implementation of a Q-learning crowd ensemble is described in greater detail in Algorithm 1. In our experiments no new hyper-parameter search is conducted when applying a crowd ensemble: the parameters found via a manual search for the baseline approaches are also used for the crowd ensemble members.

## 4 EXPERIMENTS

The purpose of the cart-pole task is to start a trial with the agent's pole in the down position and allow the agent to move the cart back and forth along a 2-D track in order to swing the pole up and balance it. The track is of finite length and each end of the track has a wall with which the cart interacts via elastic collisions. The cart-pole task has four state variables: cart position, cart velocity, pole angle, and pole angular velocity. The actions are discrete with $a \in \{-1, 0, 1\}$ which translate to push left, no push, and push right. The agent is rewarded a negative one when pointing downward, a positive one when pointing upward, and zero elsewhere.

We approximate a Q-function using an ANN with a single hidden layer with 20 nodes. The ANN inputs are the four state dimensions and scalar action value. The ANN output is the associated $Q(s, a)$ value. The parameters are updated using scaled conjugate gradient, Møller (1993). We store all experiences and sample them in batches of 1000 for each update. We update the parameters using five batches every 1000 time steps during training. The training simulation is run continuously and is never reset. Evaluation is performed using a separate simulation which is reset for each evaluation.

We also evaluate using a high-dimension state representation of the cart-pole task where the state is represented by two consecutive frames of an image of the cart-pole environment. The approach is similar to the DQN approach described in Mnih et al. (2015). The ANN has two convolutional and two fully-connected layers. No batch normalization layers are used. The convolutional layers use ReLU activation functions while the fully-connected layers use tanh. The convolutional layers have 20 and 40 features, respectively. The window size of the first convolutional layer is $6 \times 6$ and the second is $4 \times 4$. The strides for both layers is $2 \times 2$ meaning a new window starts every two pixels in both directions leading to overlap of the windows. The fully-connected layers are of size 100 and 20, respectively. The parameters are updated using ADAM, Kingma & Ba (2015).

The DQN approach requires a large number of parameters so we share the convolutional layers between the ensemble members. This is done by accumulating the gradient from the fully-connected layers of all ensemble members and using it update the shared convolutional layers.

When applying the crowd ensemble via DQN the same cart-pole simulation is used by the inputs are two sequential frames of the simulation. These can be seen in Section 8.3 along with some example features extracted by the CNN.

We also apply the crowd ensemble approach to the continuous state-action bipedal walker task, team (2016), via the DDPG algorithm which is a actor-critic approach, (Sutton & Barto, 1998, 69). The bipedal walker task objective is to train an agent to move a simple bipedal robot across a two-dimension, set-width, gently sloping, plane. The robot consists of two legs and an oblong hull which sits on top of the legs. The action space is continuous in four dimensions: the actuations for the hip and knee in each leg of the robot. The state space is represented in 24 dimensions: hull angle, hull angular velocity, hull x velocity, hull y velocity, leg one hip angle, leg one hip speed, leg one knee angle, leg one knee velocity, leg one ground contact indicator (boolean), leg two hip angle, leg two hip speed, leg two knee angle, leg two knee velocity, leg two ground contact indicator (boolean), and ten lidar measurements measuring the distance to the ground from the center of the hull from ten different angles. The reward function is designed to reward forward motion with minimal motor actuation while encouraging the agent to keep the hull from pointing downward and severely penalizing the agent if the hull touches the ground. The task is considered solved if the entire course is traversed within the allocated amount of time with a total reward greater than 300.

The parameters used for training the actor and critic ANNs for the bipedal walker were taken from the supplementary material of Lillicrap et al. (2016). DDPG views the actor outputs as defining the mean of a Normal distribution with unit variance. Combining the output of multiple actors using averaging will, most likely, result in a location of exceedingly small probability. Furthermore actor outputs will be multi-modal in four dimensions which will result in a challengingly large number of modes for which no straight-forward method to find the highest-probability location exists. Instead the crowd ensemble approach is applied to DDPG by training a single actor from the combined output of a crowd ensemble of critics.

## 5 RESULTS

The cart-pole tasks benefits from the crowd ensemble approach as shown in Figure 1 which shows that the mean reward during evaluation is improved when using any size ensemble with no significant improvement after $N_e = 5$ where $N_e$ is the ensemble size. The improved stability is evident in Figure 2 which shows four randomly-selected runs of the ensemble with $N_e = 50$ and four non-ensemble run. All selected non-ensemble runs show catastrophic forgetting while this happens in none of the ensemble examples.

Figure 1b shows that the ensemble approach solves the task earlier and more reliably than the non-ensemble approach. In this figure, all crowd ensemble agents solve the task within $1.7 \times 10^5$ time steps while 29 of 30 base Q-learners solve the task by $3 \times 10^5$.

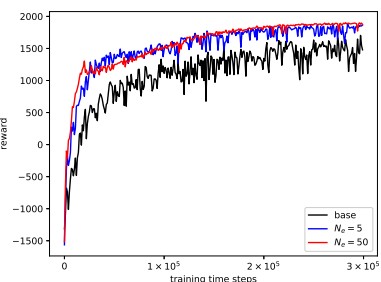

(a) Comparison of three $N_e$ values.

(b) Number of agents (out of 30) that have solved the task.

Figure 1: Comparison of crowd ensemble and standard Q-learning approaches on the low-dimension cart-pole task.

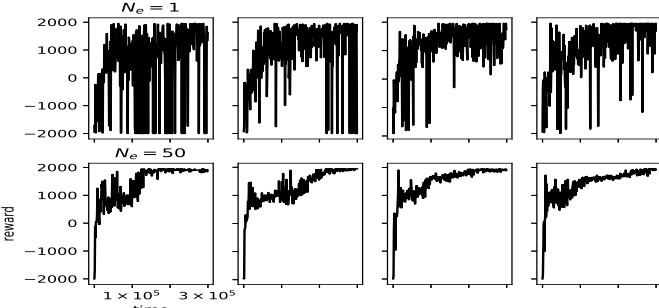

Figure 2: Randomly selected agents for $N_e = 1$, the single Q-learner, and $N_e = 50$.

Figure 3 shows the results for the high-dimension cart-pole task. It appears that the mean reward of the single DQN method will never achieve a mean performance equal to even a small crowd ensemble size. We also see continued improvement as $N_e$ increases.

Figure 4b shows the percentage of the number of agents which solved the cart-pole from pixels task for selected $N_e$ values. In this case, an evaluated reward greater than 1800 is considered solved. The crowd ensemble with $N_e = 50$ is the only one that solves the task every time within $1.5 \times 10^6$ time steps. Furthermore, all of the $N_e = 50$ agents have solved the task before a single $N_e = 1$ agent. '

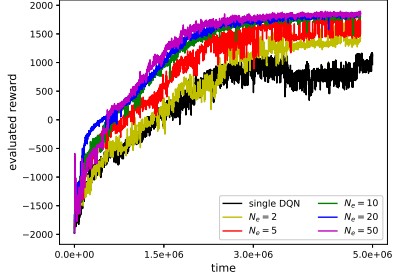
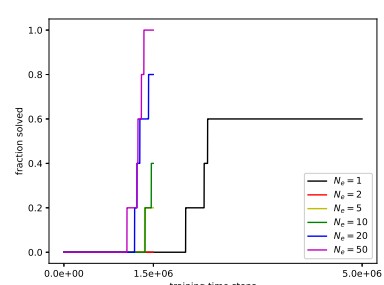

(a) Comparison of $N_e$ values against non-ensemble DQN approach.

(b) Percentage of agents that have solved the task.

Figure 3: Comparison of DQN results against the crowd ensemble DQN approach on the high-dimension cart-pole task.

Finally we look at results from the bipedal walker task in Figure 4 where we see the same benefits to the crowd ensemble as with the previous tasks. We also examine the the received reward for each

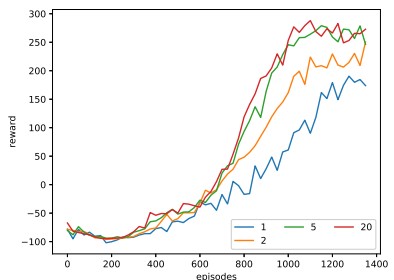

(a) Comparison of $N_e$ values against non-ensemble DQN approach.

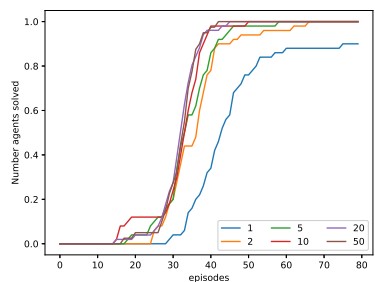

(b) Parentage of agents that have solved the task.

Figure 4: Comparison of results on the bipedal walker task.

unit of distance traveled as a measure of locomotion efficiency in Figure 5. Here we see that, as $N_e$ increases, so does the efficiency which indicates higher quality solutions. We conjecture that the improved critic stability of the crowd ensemble allows the actor to improve upon what it has learned and further refine the policy. Figure 5 shows that, as $N_e$ increases, the efficiency of the policy increases as well. It shows a histogram for each of five selected $N_e$ values. The x-axes represent the amount of reward received after every time step. All 2000 evaluation time steps for ten randomly selected solutions (agents able to traverse the entire space within the allotted time). Only solutions found after 1000 training episodes were used in order to give at the agents ample opportunity to refine their solutions.

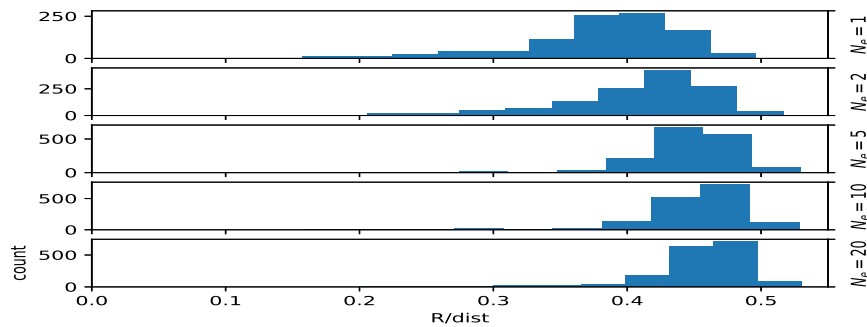

Figure 5: Solution efficiency histograms for the non-ensemble approach and the first five $N_e$ values.

# 6 DISCUSSION: CROWD ENSEMBLES STABILIZE THE DECISION SPACE

While Q-functions, $Q(s, a)$, represent the agent's knowledge about how to solve a task, it is the decision space, $\arg\max_a Q(s, a)$, that is the application of this knowledge. Figure 6 shows both for two of the dimensions of the low-dimension cart-pole task.

The decision space is dominated by a boundary between the push left and push right decisions which runs diagonally through this 2-dimension cross-section of the space. The Q-function is dominated by a ridge of high Q-values which also runs diagonally across the center of the decision boundary.

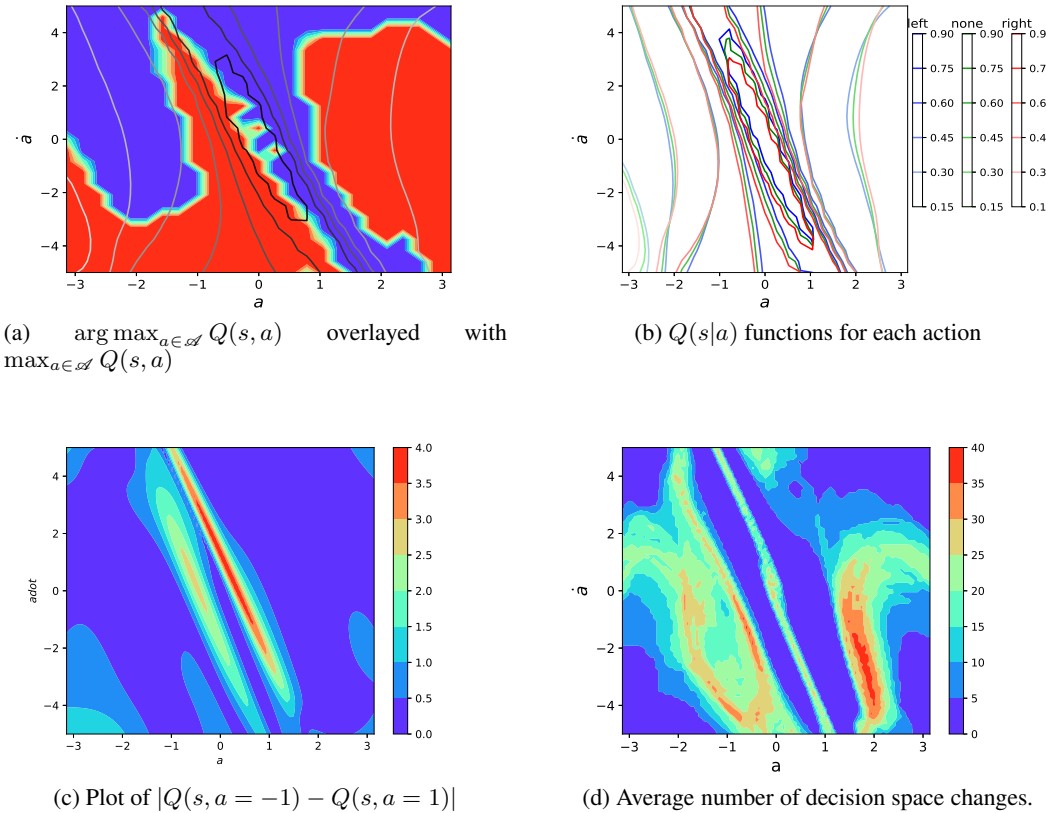

(a) $\arg\max_{a\in\mathscr{A}} Q(s,a)$ overlayed with $\max_{a\in\mathscr{A}} Q(s,a)$

(b) $Q(s|a)$ functions for each action

(c) Plot of $|Q(s, a = -1) - Q(s, a = 1)|$

(d) Average number of decision space changes.

Figure 6: Two-dimension views of the Q-function and decision space volatility for the low-dimension cart-pole task for a non-ensemble Q-function.

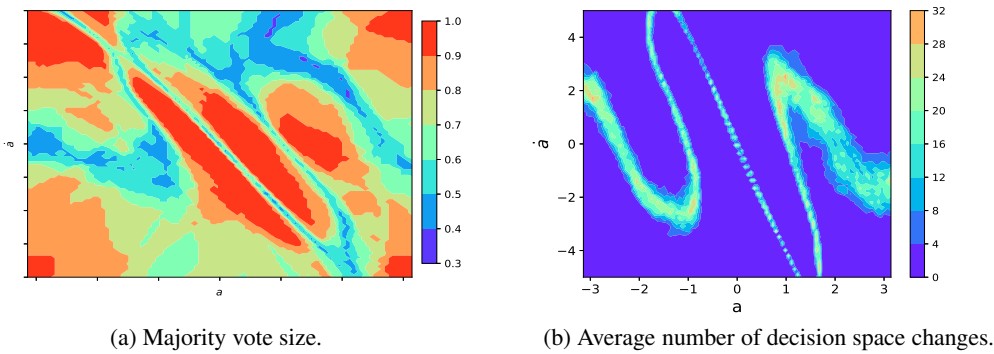

(a) Majority vote size.

(b) Average number of decision space changes.

Figure 7: Two-dimension views of the Q-function and decision space volatility for the low-dimension cart-pole task for a Q-function ensemble of size 20.

The Q-function can be plotted for each of the discrete actions as shown in Figure 6b. Here we see that there is a sizable difference in the Q-function across state positions but that the Q-values are only slightly shifted across the actions.

This results in large differences in $Q(s, a = -1) - Q(s, a = 1)$ in the steepest regions of the Q-function but relatively small differences in the state space regions with less dramatic relief. This relationship is visible in Figure 6c where the Q-function regions with greatest rate of change have the largest differences in Q-functions across actions.

What effect does this have? Every parameter update will result in changes to $Q(s, a)$ over the entire state space. Regions of the state space with small differences in Q-function across actions will be susceptible to spurious changes in the decision space than those with large differences. This is evident in Figure 6d which is a count of the average number of decision space changes (i. e., the number of times $\arg\max_a Q(s, a)$ changed) over ten reruns for a non-ensemble Q-learner after each agent has already solved the task and after $1 \times 10^5$ training time steps have occurred. Here we see that the number of decision space changes during training is nearly zero in the high Q-difference regions. These regions of high decision space volatility are the culprit behind instability during training because the agent must pass through them on its way to state space regions with greater stability. Additional views of the state space can be found in Section 8.2.

The crowd ensemble has a stabilizing effect on the decision space. For a single Q-function the majority of parameter updates result in improvements toward or sustaining a decision space which successfully leads to the goal state. However a significant minority of the updates are destructive enough to cause significant changes in the decision space which will lead to forgetting. Therefore a simple combination of the ensemble members' decisions will be much more likely to avoid destabilizing changes to the decision space. Figure 7b shows the number of action changes for an ensemble of size $N_e = 20$ which are reduced compared to the changes in Figure 6d.

This improved stability comes from the relatively large barrier to change of a decision space which is based upon majority voting. The average majority size across ten ensembles of size twenty is shown in Figure 7a. Unlike the small Q-differences which are frequently overcome by the noisy path taken the by parameter updates, the majority sizes are not so easily overcome.

## 7 Conclusions and future work

We have presented a simple ensemble approach to Q-learning which confronts the most common complaints about RL: that training takes too long and that training is unstable. The crowd ensemble approach to Q-learning can be used in tandem with other approaches including the most recent advances in the field. Instead of multiplying the number of interactions with the environment, as in recent, high-profile work, it increases the computational requirements but reduces the number of required interactions with the environment. Furthermore, each member of the crowd ensemble can be trained in parallel allowing for a negligible increase in wall-clock time. Our experiments demonstrate that the approach improves performance by reducing decision space volatility resulting in improved mean reward, a near elimination of catastrophic forgetting, an increase in the speed and reliability of learning, and an improvement in the quality of the solutions.

Our results regarding shared convolutional layers point to the potential for a dramatic speed-up in training for domains with high-dimension inputs. In this instance the instability of Q-learning worked to our advantage by providing the shared layers with a combined gradient which leads to more direct path toward high-quality features.

An important item of remaining work is a comparison of the crowd ensemble against more traditional ensemble methods such as mixtures of experts which have not received widespread adoption in the RL literature. An important limitation of the crowd ensemble approach to Q-learning is that the ability of the ensemble is limited by the ability of the ensemble members. Mixture of experts style ensembles are designed specifically to not have this problem.

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

## 8 APPENDIX

### 8.1 PSEUDO CODE ALGORITHM OF CROWD ENSEMBLE Q-LEARNING

An implementation of a crowd ensemble is shown in Algorithm 1. Specifically, this algorithm describes the steps used to train an ensemble of Q-functions to solve the low-dimension cart-pole task.

Lines 13 and 14 are the action voting and action selection steps. Lines 22–27 are repeated for each ensemble member meaning each ensemble member draws a unique set of training sequences to replay during training. Lines 24 and 25 are computations of the member-specific Q-function target values and error function for use during parameter updates.

---

**Algorithm 1:** Pseudo-code of the crowd ensemble for the low-dimension cart-pole Q-learning implementation using a two-layer ANN to model the Q-function.

1   Initialize ANN weights;
2   $t \leftarrow 0$;
3   **while** $t < T$ **do**
4       Obtain initial $s(t_B = 0)$, $a(t_B = 0)$, $r(t_B = 0)$;
5       $t_B \leftarrow 1$;
6       **while** $t_B \leq N_B$ **do**
7          $s(t_B) \leftarrow \texttt{Act}\,(a(t_B - 1))$;
8          $r(t_B) \leftarrow \texttt{Reward}\,(s(t_B))$;
9          **if** $\texttt{Random}\,(0,1) > \epsilon$ **then**
10             Set $a(t_B)$ to random action;
11          **end**
12          **else**
13             $v_a(t) \leftarrow \sum\limits_{c=1}^{N_e} \delta\big(\arg\max_{a'} Q^{(c)}(s(t), a'), a\big)$;
14             $a(t) \leftarrow \arg\max_a v_a(t)$;
15          **end**
16          $S_b(t) \leftarrow \big(s(t_B), r(t_B), a(t_B)\big)$;
17          $t_B \leftarrow t_B + 1$;
18          $t \leftarrow t + 1$;
19       **end**
20       Store training sequence, $S_b$, of length $N_B$ in memory;
21       **for** $c \in \{1 \ldots N_e\}$ **do**
22          **for** $t_R \in \{1 \ldots N_R\}$ **do**
23             Randomly draw a training episode sequence, $S_u$, from memory;
24             $T_V^{(c)}(t_B) \leftarrow r(t_B) + \gamma * \max_a Q^{(c)}(s(t_B), a) \; \forall t_B \in \{1, \ldots, |S_u|\}$;
25             $\delta_o^{(c)} \leftarrow \sum\limits_{t_B=1}^{N_B} T_V^{(c)}(t_B) - Q^{(c)}(s(t_B), a(t_B))$;
26             Perform 5 updates of ANN weights using SCG;
27          **end**
28       **end**
29       **if** $t \,\%\, evalFrequency$ **then**
30          Evaluate by performing lines 6–18 and setting $\epsilon \leftarrow 0$.
31       **end**
32   **end**

---

## 8.2 DECISION SURFACES CHANGES IN FREQUENTLY VISITED STATE SPACES

Here we provide frames from two movies showing decision space volatility in state spaces visited for a typical solution to the cart-pole task. Surfaces changes across 20 non-ensemble Q-learners once each had solved the task and continuing until training stopped. The figure shows the fraction of those parameter updates which resulted in a decision space change at that state location. Here we see that an agent must pass through several regions of high volatility on its way to the goal. Once the goal is reached, however, it is safely located between two relatively unchanging regions. These figures show that, in the cart-pole swing-up task, the greatest cause of catastrophic forgetting where the agent appears to have forgotten much of what it has learned is a result of volatility in the regions of the state space that must be visited on the way to the goal.

Figure 9 shows the same states as Figure 8 but the decision space changes are computed from an ensemble of size $N_e = 20$. The plots are scaled to keep the colors consistent between figures. In fact, the maximum fraction of time steps that a particular location in state space changes its selected action is nearly identical for the $N_e = 20$ and $N_e = 1$ case: just over $47\%$ of the parameter updates.

The difference in state space volatility is striking! The ensemble has not removed volatility but it has mitigated it considerably. Furthermore, the regions of Figure 9 which have the most instability are the regions where it matters least: namely the decision surface boundary in and around the goal region which is surrounded by regions of low decision space volatility. The other regions of high decision space volatility for the ensemble is when the pole is pointed downward with little angular velocity where the action decision had little impact.

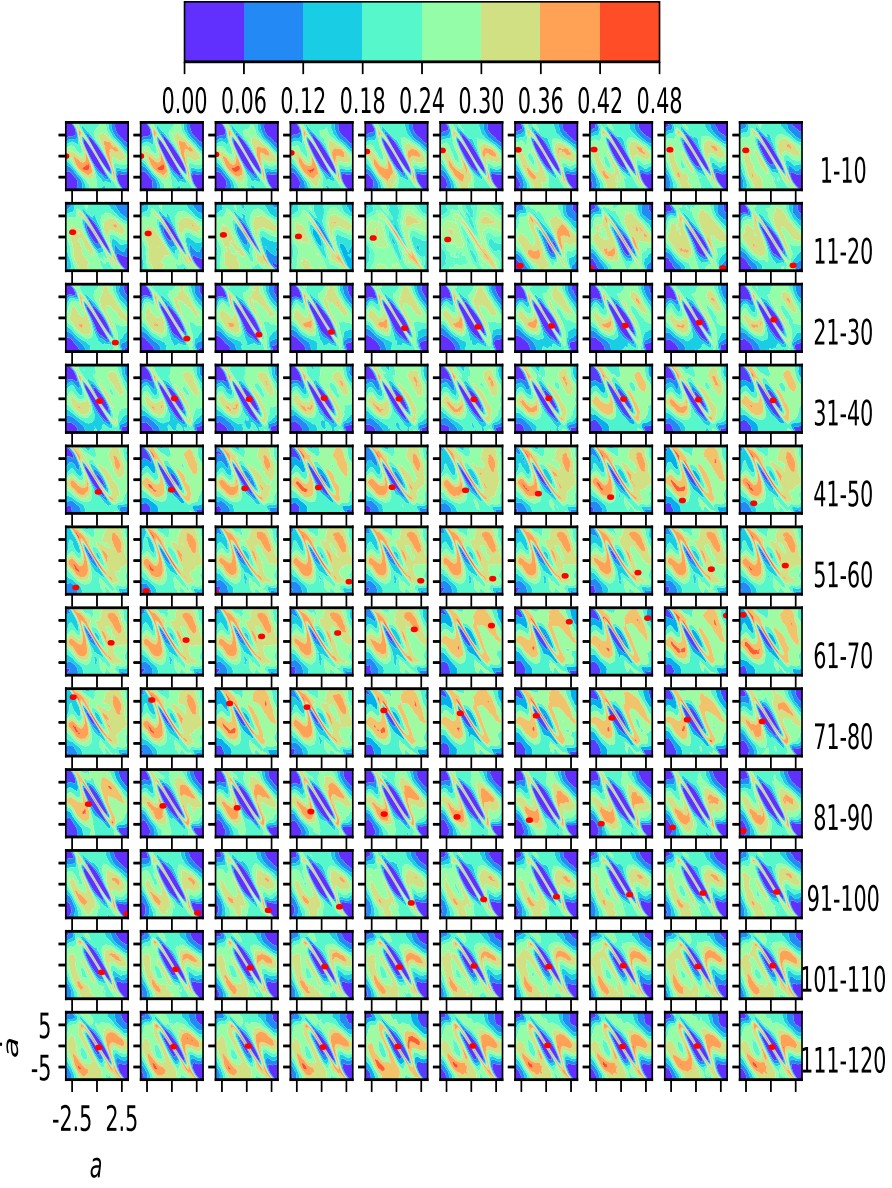

Figure 8: Frames of a movie showing the agent's movement through the state space for an example run of a single Q-learner ($N_e = 1$). Each Frame is a single time step. The first 120 time steps are shown. State space locations are colored according to the number of times the preferred action at that location changed. The red dot in the agent's current location in state space. Color bar is scaled to match colors in Figure 9

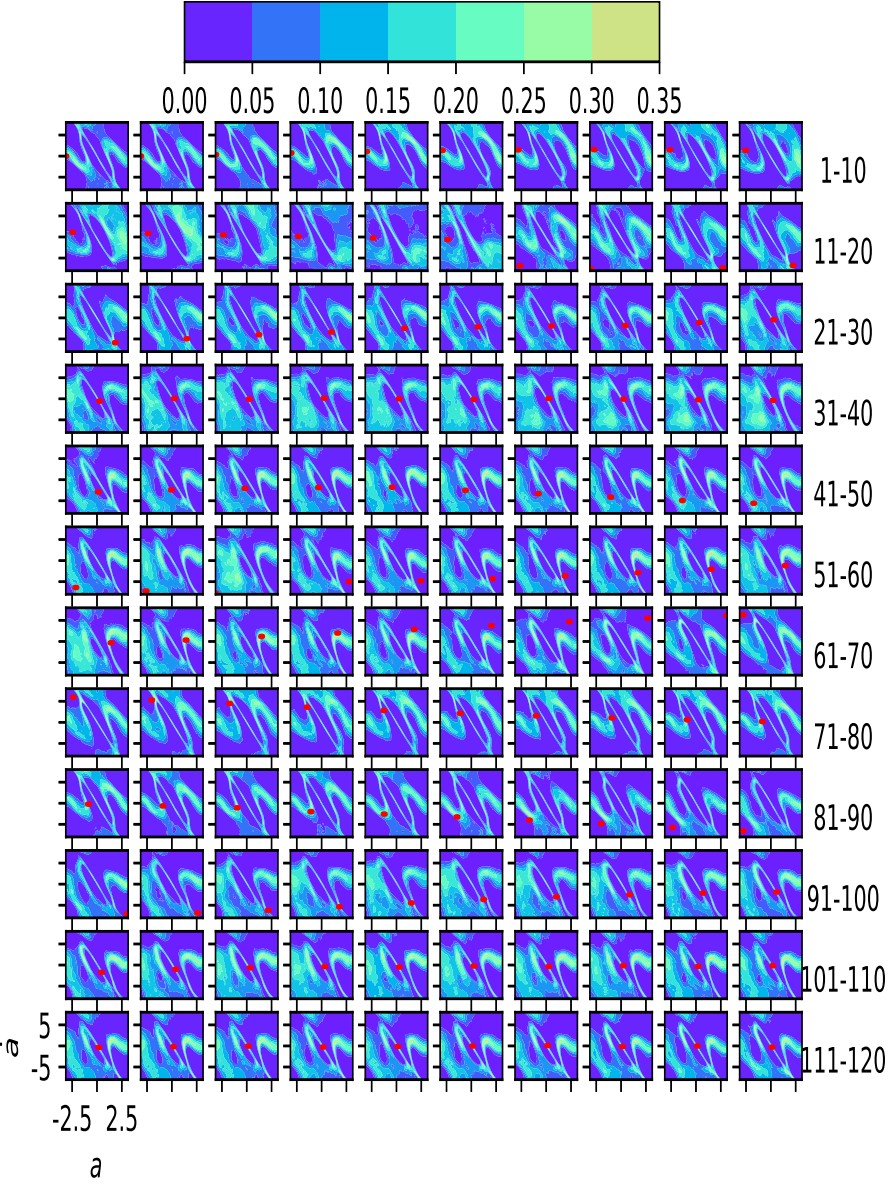

Figure 9: Frames of a movie showing the agent's movement through the state space for an example run of an ensemble Q-learner ($N_e = 20$). Each Frame is a single time step. The first 120 time steps are shown. State space locations are colored according to the number of times the preferred action at that location changed. The red dot in the agent's current location in state space. Color bar is scaled to match colors in Figure 8

8.3 CROWD ENSEMBLE DQN ADDITIONAL INFORMATION

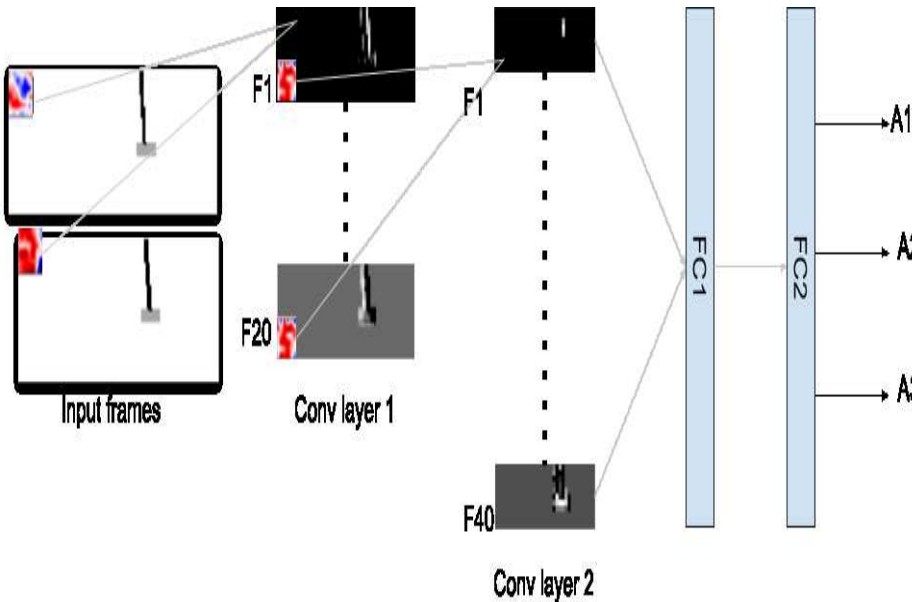

Figure 10: Example deep Q-learning network with two convolutional layers and two fully-connected layers and three action outputs. Shown are two example input frames and two example features at each convolutional layer. The input frames and subsequent features were captured from our DQN agent during training.

Figure 10 shows the output of two of the features each from two convolutional layers computed from the two input frames. In these frames the pole is rotating clockwise. Feature one appears to highlight the direction of the cart and pole which are both rightward in this example. The bright white pixels of feature one of layer one appear to be the leading edge of the cart and pole indicating the direction. Feature 20 of the first layer may encode the opposite information. Feature one of the second layers appears to encode the location of the pole while feature 40 may encode the location of the cart.

