# OpenReview forum: "The wisdom of the crowd: reliable deep reinforcement learning through ensembles of Q-functions"
_ICLR.cc/2019/Conference_

### Official Review · AnonReviewer3 · 2018-10-30
**review report**

**Rating:** 3
**Confidence:** 4

**Review:**

This paper introduces an ensemble version of Deep RL by bagging Q-function approximation estimates. In the experiments, the performance of the proposed work is compared to the baseline, single DQN. In spite of the contribution, this paper has a critical issue.

It has been extensively studied in the literature that ensemble DQN could lead to better performance than a single DQN. See the seminal work by Osband et al. (2016). The authors did not cite this paper, not to say a long list of recent works who have cited this seminal work. This indicates that the authors fail to conduct a serious literature review. In addition, more comprehensive experiments are required to compare the proposed work with the state-of-the-art ensemble DQN methods.

Osband et al. (2016), Deep Exploration via Bootstrapped DQN. NIPS.

---

### Official Review · AnonReviewer2 · 2018-11-03
**Interesting idea while the experiments are not enough.**

**Rating:** 5
**Confidence:** 3

**Review:**

This paper proposes the deep reinforcement learning with ensembles of Q-functions. Its main idea is updating multiple Q-functions, instead of one, with independently sampled experience replay memory, then take the action selected by the ensemble. Experimental results demonstrate that the proposed method can achieve better performance than non-ensemble one under the same training steps, and the decision space can also be stabilized.

This paper is well-written. The main ideas and claims are clearly expressed. Using ensembles of Q-function can naturally reduce the variance of decisions, so it can speed up the training procedure for certain tasks. This idea is simple and works well. The main contribution is it provides a way to reduce the number of interactions with the environment. My main concern about the paper is the time cost. Since the method requires updating multiple Q-functions, it may cost much more time for each RL time step, so I’m not sure whether the ensemble method can outperform the non-ensemble one within the same time period. This problem is important for practical usage. However, the authors didn’t show these results in the paper.

Minor things:
+The main idea is described too sketchily. I think more examples, such as in section 8.1, should be put in the main text.
+Page6 Line2, duplicated ‘the’.

---

### Official Review · AnonReviewer1 · 2018-11-08
**Not enough novelty**

**Rating:** 4
**Confidence:** 5

**Review:**

This paper proposes a cute idea as suggesting ensembles of Q-function approximations rather than a singular DQN.

However, at the core of it, this boils down to previously studied methods in the literature, one of which also is not cited here:

@inproceedings{osband2016deep,
  title={Deep exploration via bootstrapped DQN},
  author={Osband, Ian and Blundell, Charles and Pritzel, Alexander and Van Roy, Benjamin},
  booktitle={Advances in neural information processing systems},
  pages={4026--4034},
  year={2016}
}

Experiments provided in this paper compares with only the weak baseline of single DQN, however, it fails to compare other similar ideas in the literature such as the above paper. Hence, this paper lacks enough novelty for publication, and it is not clear from the experiments that the specific method proposed in this paper is better than others in the SOTA.

---

### Meta-Review · Area_Chair1 · 2018-12-14
**The paper can be improved**

**Confidence:** 4
**Recommendation:** Reject

**Metareview:**

The paper suggests using an ensemble of Q functions for Q-learning. This idea is related to bootstrapped DQN and more recent work on distributional RL and quantile regression in RL. Given the similarity, a comparison against these approaches (or a subset of those) is necessary. The experiments are limited to very simple environment (e.g. swing-up and cart-pole). The paper in its current form does not pass the bar for acceptance at ICLR.